# Anticancer Potential of Temozolomide-Loaded Eudragit-Chitosan Coated Selenium Nanoparticles: In Vitro Evaluation of Cytotoxicity, Apoptosis and Gene Regulation

**DOI:** 10.3390/nano11071704

**Published:** 2021-06-29

**Authors:** Madineh Mazarei, Pooria Mohammadi Arvejeh, M. R. Mozafari, Pegah Khosravian, Sorayya Ghasemi

**Affiliations:** 1Cellular and Molecular Research Center, Basic Health Sciences Institute, Shahrekord University of Medical Sciences, Shahrekord 88157-13471, Iran; n.nazanin1252@gmail.com (M.M.); pooriam9473@gmail.com (P.M.A.); 2Supreme NanoBiotics Co. Ltd. and Supreme Pharmatech Co. Ltd., 399/90-95 Moo 13 Kingkaew Rd. Soi 25/1, T. Rachateva, A. Bangplee, Samutprakan 10540, Thailand; dr.m.r.mozafari@gmail.com; 3Australasian Nanoscience and Nanotechnology Initiative (ANNI), Monash University LPO, Clayton 3168, Australia; 4Medical Plants Research Center, Basic Health Sciences Institute, Shahrekord University of Medical Sciences, Shahrekord 88157-13471, Iran

**Keywords:** glioblastoma multiforme, temozolomide, selenium nanoparticle, Cs, Eud, RELA, MGMT, E2F6

## Abstract

Resistance to temozolomide (TMZ) is the main cause of death in glioblastoma multiforme (GBM). The use of nanocarriers for drug delivery applications is one of the known approaches to overcome drug resistance. This study aimed to investigate the possible effect of selenium–chitosan nanoparticles loaded with TMZ on the efficacy of TMZ on the expression of MGMT, E2F6, and RELA genes and the rate of apoptosis in the C6 cell line. Selenium nanoparticles (SNPs) were loaded with TMZ and then they were coated by Eudragit^®^ RS100 (Eud) and chitosan (C_S_) to prepare Se@TMZ/Eud-Cs. Physicochemical properties were determined by scanning electron microscope (SEM), energy-dispersive X-ray spectroscopy (EDAX), thermal gravimetric analysis (TGA), Fourier-transform infrared spectroscopy (FTIR), and dynamic light scattering (DLS) methods. Se@TMZ/Eud-Cs was evaluated for loading and release of TMZ by spectrophotometric method. Subsequently, SNPs loaded with curcumin (as a fluorophore) were analyzed for in vitro uptake by C6 cells. Cytotoxicity and apoptosis assay were measured by MTT assay and Annexin-PI methods. Finally, real-time PCR was utilized to determine the expression of MGMT, E2F6, and RELA genes. Se@TMZ/Eud-Cs was prepared with an average size of 200 nm as confirmed by the DLS and microscopical methods. Se@TMZ/Eud-Cs presented 82.77 ± 5.30 loading efficiency with slow and pH-sensitive release kinetics. SNPs loaded with curcumin showed a better uptake performance by C6 cells compared with free curcumin (*p*-value < 0.01). Coated nanoparticles loaded with TMZ showed higher cytotoxicity, apoptosis (*p*-value < 0.0001), and down-regulation of MGMT, E2F6, and RELA and lower IC50 value (*p*-value < 0.0001) than free TMZ and control (*p*-value < 0.0001) groups. Using Cs as a targeting agent in Se@TMZ/Eud-Cs system improved the possibility for targeted drug delivery to C6 cells. This drug delivery system enhanced the apoptosis rate and decreased the expression of genes related to TMZ resistance. In conclusion, Se@TMZ/Eud-Cs may be an option for the enhancement of TMZ efficiency in GBM treatment.

## 1. Introduction

Glioblastoma multiforme (GBM) is the most common, aggressive, and deadly primary brain tumor in adults, with a median survival of 12–15 months post diagnosis [1]. The main cause of mortality in GBM patients is excessive drug resistance [2]. Although temozolomide (TMZ) is an alkylating chemotherapy key agent for GBM treatment, there is considerable resistance to this drug, which causes recurrence of the disease [3]. The anti-cancer mechanism of TMZ is as follows: cell cycle suppression in the G2/M stage, DNA damage, and consequently, GBM cells apoptosis [3,4]. This process ensues during DNA replication by carrying a methyl group that attaches to guanine at the O6 and N7 positions, and adenine at the N3 position. These mismatched lethal base pairs can result in single- and double-strand DNA breaks (DSB) that lead to cell apoptosis [5,6]. TMZ is a class of imidazotetrazines [7,8,9] that is an alkylating agent, which is stable at acidic pH but is rapidly converted to 3- methyl-(traiazan-1-yl) imidazole- 4- carboxamide (MTIC) at physiological pH [7,8]. The MTIC methylates DNA, but it is not able to cross the blood–brain barrier (BBB) and displays a low cell absorption, thus diminishing healing efficacy [10]. Furthermore, the molecular mechanisms in TMZ resistance have been identified. One of the most significant cellular factors involved in TMZ resistance therapy is MGMT (O-6-Methylguanine-DNA Methyltransferase) [11,12]. It is a 22 kD DNA repair enzyme that destroys the methyl group in DNA. Therefore, MGMT is involved in chemotherapy resistance due to its ability to eliminate the toxicity of alkylating agents, such as TMZ. MGMT promoter methylation was associated with enhanced patients’ survival when they were cured with alkylating agents like TMZ [13]. E2F6 is another key factor in TMZ resistance in GBM patients. E2F6 enhances DSB repairability; thus, overexpression of E2F6 increases the TMZ resistance [4]. EGFRvIII/AKT/NF-κB signaling pathway controls the E2F6 expression level that is a direct target gene of NF-κB [14]. Binding the NF-κB complex to the MGMT promoter leads to TMZ chemoresistance. NF-κB expression is excessive in glioma tissues and inversely related to the overall survival (OS) of patients [15]. On the other hand, RELA gene expresses p65 (RelA) which is a subunit of the NF-κB family. GBM has a high NF-κB expression and affects many of the major GBM oncogenic pathways. TMZ therapy can significantly activate the NF-κB signaling pathway in GBM [16]. It is shown that NF-κB inhibitor with TMZ can create significant apoptosis in GBM via the p65 (RelA) protein [17]. Several strategies have been proposed to overcome resistance to cancer chemotherapies. Nanocarriers have the potential to limit drug toxicity, achieve tumor localization, and overcome drug resistance [18,19]. On the other hand, brain drug delivery and passing the blood–brain barrier (BBB) are directly affected by particle size. Therefore, the size of nanoparticles also plays a role in effective treatment [20,21]. In cancer treatment, selenium compounds can be used to overcome drug resistance and recurrent cancer [22]. Moreover, selenium nanoparticles (SNPs) display synergistic effects with their therapeutic cargo and improve anticancer activity [23,24].

In this research we aimed to formulate a multicomponent, selenium nanoparticle-based therapeutic for GBM patients. In order to achieve optimum efficacy and site-specific drug delivery we employed chitosan and Eudragit^®^ RS100 (Eud) in the formulation. Chitosan (Cs) is a natural, non-toxic, cationic polymer with some favorable mucoadhesive and anticancer properties, including the ability to induce apoptosis and targeted drug delivery to tumors [25,26]. It is proven that the use of Cs enhances TMZ stability by loading in Cs-based nanoparticles [10]. Eud, on the other hand, is a pH-dependent polymer that can be dissolved in a medium with pH > 5.5 [27,28]. Since overexpression of MGMT, E2F6, and RELA (NF-κB) genes with well-known mechanisms are involved in TMZ drug resistance, the purpose of this study was to design and prepare an SNP coated with Eud-Cs for TMZ delivery to a TMZ-resistant cell line (C6 cell line) [29]. The designations used throughout the manuscript for the designed formulations are depicted in Figure 1. After characterization and evaluation of the unloaded (empty) nanoparticles and nanodrug particles, the effect of further delivery of TMZ was measured via expression levels of MGMT, E2F6, and RELA genes and cell apoptosis compared to the control samples with no nanocarriers.

## 2. Materials and Methods

### 2.1. Reagents

The C6 cell line (rat GBM) was obtained from the Pasteur Institute of Iran (Tehran, Iran). Dulbecco’s Modified Eagle’s Medium with 1000 mg/mL glucose (DMEMF12), fetal bovine serum (FBS), penicillin-streptomycin solution (10,000 units/mL penicillin and 10 mg/mL streptomycin in normal saline), phosphate buffered saline (PBS; pH 7.4), and trypsin-EDTA (0.25% trypsin, 1 mM) were purchased from Biosera (Vienna, Austria). Commercial Eud (Eudragit^®^ RS100) was obtained from Evonik industries (Essen, Rhine–Westphalia, Germany). TMZ, low molecular Cs, and other reagents and solvents were purchased from Merck (Darmstadt, Hesse, Germany). 

### 2.2. Nanoparticle Preparation

Initially, 55.5 mg of selenium dioxide was dissolved in 500 mL of deionized water by stirring. Then, 387.2 mg of ascorbic acid was dissolved in 50 mL of deionized water and it was added dropwise to the first solution with a stirring speed of 500 rpm [3]. During the addition of the ascorbic acid solution, SNPs were formed, and the color of the solution turned red. The red color suspension of SNPs was spun for about 1 h. In the following step, 10 mL of the suspension was added to 1 L of deionized water and 50 mL of TMZ solution (5 mg/mL in deionized water) was added and stirred for 2 h. Next, Eud solution (20 mg in 10 mL of acetone) was subjoined gradually to the last suspension and it was stirred for 24 h at 4 °C [30]. The sediment was separated by centrifugation at 20,000 rpm for a period of 30 min. In this step, Se@TMZ/Eud nanoparticles were prepared. In order to synthesize Se@TMZ/Eud-Cs nanoparticles, 5 mL of the Cs solution (5 mg in 0.1 M acetic acid) was added to the Se@TMZ/Eud suspension (before the centrifuge step) and placed on the stirrer for 12 h [31]. The final Se@TMZ/Eud-Cs nanoparticles were obtained as explained in the previous centrifugation step. Prepared nanoparticles were dried by freeze-drying instrument and used for further analysis.

### 2.3. Characterization of the Nanoparticles

Several techniques were employed to characterize the prepared nanoparticles. Se, Se@TMZ/Eud, and Se@TMZ/Eud-Cs were observed under the scanning electron microscopy (FE-SEM, Tescan/Mira, Brno, Czech Republic) to evaluate the shape and surface morphology of the nanoparticles. Moreover, the particle size, size distribution, and zeta potential of the nanoparticles were assessed by DLS (Mastersizer 2000; Malvern Instruments, Malvern, Worcestershire, UK). To investigate the surface elements of the final nanoparticle, the EDAX method was used with a field-emission scanning electron microscope (FE-SEM, Tescan/Mira, Brno, Czech Republic) coupled with an EDAX detector. The surface modification of the nanoparticles in each step was studied by infrared (IR) spectroscopy recorded on a Nicolet Magna IR-550 (Madison WI, USA) using KBr pellets. Thermal gravimetric analysis (TGA) system (STA PT 1600, Linseis, Germany) with a temperature ramp of 5 °C/min or 10 °C/min was employed under N_2_ atmosphere with a rate of 50 mL/min to determine TGA curves of the SNPs. 

### 2.4. Evaluation of Drug Content

To evaluate the loaded TMZ, an indirect method was used. For this, at the centrifuge step of nanoparticle preparation, the supernatant was separated and evaluated. TMZ content was measured by a UV–Vis spectrophotometer at 326 nm with using a calibration curve [32]. TMZ loading efficiency and TMZ loading capacity of the nanoparticles were calculated employing Equations (1) and (2), respectively: % TMZ Loading efficiency = (Initial amount of TMZ − Amount of TMZ in supernatant)**/**(Initial amount of TMZ) × 100(1)
% TMZ Loading capacity = (Initial amount of TMZ − Amount of TMZ in supernatant)**/**(Amount of final nanoparticles) × 100(2)

### 2.5. Evaluation of Drug Release

Under sink condition, 2 mg of TMZ-loaded nanoparticles were poured into a dialysis bag and 2 mL phosphate-buffered saline (PBS) was added to it. The closed bag was placed in a glass beaker containing 20 mL PBS to set in a shaker incubator at 37 °C and 100 rpm. At this stage, two BPS solutions with pH 7.4 and pH 5.0 were used. At the preschedule sampling time, 1 mL sample was taken and 1 mL of fresh PBS was added to measure the absorbance value using a UV–Vis spectrophotometer at the wavelength of 326 nm [33,34,35].

### 2.6. Evaluation of Cellular Uptake of Nanoparticles by Flow Cytometry

In order to evaluate the cellular uptake efficiency of Se@TMZ/Eud-Cs, curcumin (Cur) molecule, which has intrinsic fluorescent properties, was used to label SNPs [31]. For the synthesis of Se@Cur/Eud-Cs, 5 mL of Cur solution (1 mg/mL in acetone) was used instead of TMZ solution and the other steps were performed as before. To evaluate cellular uptake of nanoparticles, C6 cells were cultured in 6-well plates with a density of 5 × 10^5^ and incubated for 24 h. Subsequently, cells were treated with Cur and Se@Cur/Eud-Cs with the same concentration of Cur for 4 h. The cells were then washed with PBS to remove the culture medium. Imaging was performed by flow cytometry (flow cytometer Partec, Space Xenon/Argon laser, Sysmex Partec GmbH, Münster, Germany).

### 2.7. Cell Culture and Cell Viability Assay

C6 cell line (rat GBM) was stored in −70 °C. C6 cells were cultured in Dulbecco’s modified Eagle’s medium (DMEM F12) supplemented with 10% fetal bovine serum (FBS), 100 IU/mL streptomycin, and 100 IU/mL penicillin, in an incubator at 37 °C, 5% CO_2_, and 95% humidity. Cell passaging was done after 48 h using trypsin-EDTA (0.25%). Cell viability was evaluated by the MTT assay for the Se@TMZ/Eud-Cs and Se@TMZ/Eud samples in comparison with free TMZ and control. Cells were seeded at a frequency of 5000 cells per well (96-well plate). They adhered to the bottom of the plate in an incubator at 37 °C after 24 h. Then, 200 µL of different concentrations of TMZ (0–1200 µL) and TMZ-loaded SNPs with the same concentration of TMZ were treated for 48 h. In the next step, 20 µL MTT (5 mg/mL) was added to each well. Cells were incubated for 4 h at 37 °C. To remove the contents of the wells, 200 µL DMSO was added to each well and incubated for 20 min. Finally, they were centrifuged and the absorbance of the supernatant was determined at 570 nm with the ELISA reader (STAT FAX 2100, Ramsey, MN 55303, USA). 

### 2.8. Apoptosis Assay

The C6 cells (with a density of 5 × 10^5^) were seeded in a 6-well plate and were then treated with TMZ, Se@TMZ/Eud-Cs, and Se@TMZ/Eud with 200 µM concentration of TMZ for 48 h at 37 °C and 5% CO_2_. Untreated cells were used as the control group. In the following stage, the culture medium was removed and cells were washed twice with PBS. 1 mL Annexin V-FITC/PI (BD Biosciences cat: 51-66211E) was added to each well for 20 min at room temperature in darkness. Afterwards, the solution was removed after adding 1 mL PBS to the wells, and C6 cells were analyzed by flow cytometry (flow cytometer Partec, Space Xenon/Argon laser, Hamburg, Germany).

### 2.9. RNA Extraction and cDNA Synthesis

The C6 cells (with a density of 1 × 10^6^) were cultured in each of the 10 cm^2^ plates. After 24 h of seeding, cells were treated with different formulations, i.e., TMZ, Se@TMZ/Eud, and Se@TMZ/Eud-Cs (all with the same concentration of 200 μM TMZ). After 48 h, RNA was extracted with RiboEx reagent (GeneAll, #301-001) after removing the culture medium from the plates according to the manufacturer’s instructions. Cells were lysed by adding 1 mL of RiboEx reagent. After that, through washing with chloroform and ethanol (according to the manufacturer’s manual), total RNA was obtained. Total RNA concentration and quantity (OD adsorption at 260 nm) were measured using a Thermo Scientific™ NanoDrop 2000. By using gel electrophoresis, the quantity of the extracted RNA was measured. cDNA synthesis was done with 1 μg of total RNA, by utilizing a random hexamer primer and cDNA Synthesis Kit (Yekta Tajhiz Azma, Iran #YT4500) following the manufacturer’s protocol.

### 2.10. Real-Time PCR

The primers for qPCR analysis for rat MGMT, E2F6, RELA, and Actb (as internal control) genes were designed by Primer3 web-based server [36] and Integrated DNA Technologies (IDT) (Coralville, IA, USA). Their specificity was tested with Primer-BLAST of the NCBI genome browser. The used primer sequences are listed in Table 1. The mRNAs expression levels were carried out on a Rotor-Gene 3000 system (Corbett Research, Sydney, Australia), using the synthetic primers and Real Q Plus 2x Master Mix Green (Ampliqon, Denmark #A323402), according to the protocol provided by the manufacturer. Eventually, a 2^−ΔΔct^ method was used for gene expression analysis.

### 2.11. Statistical Analysis

All data were presented as mean ± SD. All statistical analyses were performed using GraphPad Prism v.8.4.3. Intergroup comparisons were made using ANOVA test followed by the Tukey method and *p* < 0.05 was considered to be significant. All cell manipulation tests were performed in triplicate.

## 3. Results

### 3.1. Characterization of Nanoparticles

#### 3.1.1. Morphological and Surface Properties

Morphological features and surface properties of prepared SNPs were investigated by the field emission scanning electronen microscope (FE-SEM) instrument. Figure 2A confirmed the spherical shape and uniform distribution of SNPs. The size of SNPs was approximately 150 nm (Figure 2A). An approximate 20 nm increase in the size of the drug-loaded particles was observed after covering the surface of SNPs with the Eudragit polymers (Figure 2B). The thickness of the coated layer increased more in the final nanoparticles when compared with those obtained in the previous step. The size of Se@TMZ/Eud-Cs particles was about 200 nm (Figure 2C).

#### 3.1.2. Evaluation by DLS and SBL

The SNPs and Se@TMZ/Eud-Cs were evaluated by the dynamic light scattering (DLS) method and sparse Bayesian learning (SBL) calculation, which assists to reduce the agglomeration error, particularly at the nanoscale. Consequently, the obtained particle size by this method is close to the results obtained by SEM. As observed in Figure 3a,b, SNPs and Se@TMZ/Eud-Cs exhibited an average diameter of 169 and 228 nm, respectively. These data are in agreement with the results obtained by the SEM studies. The sharp peaks at Figure 3a,b present a narrow size distribution for the mentioned nanoparticles. Furthermore, the zeta potential of SNPs and Se@TMZ/Eud-Cs were investigated. The zeta potential value obtained for SNP was −20 mV (Figure 3c). As Figure 3d shows, the positive zeta potential for Se@TMZ/Eud-Cs (+32 mV) demonstrated surface modifications of Se@TMZ/Eud-Cs by Cs.

#### 3.1.3. Validity Measurement by EDAX Assay

Se@TMZ/Eud-Cs were evaluated by the energy-dispersive X-ray analysis (EDAX) method to determine elemental composition (Figure 4). It was found that the elements of selenium (Se), carbon (C), oxygen (O), nitrogen (N), and chlorine (Cl) were present in final nanoparticles with the atomic ratio of 0.1, 48.4, 36.1, 13.8, and 0.7%, respectively. These data, therefore, re-assured that SNPs were coated by Cs and Eud on the surface.

#### 3.1.4. Fourier Transform Infrared Spectroscopy (FTIR) Assay

FTIR confirmed the formation of Se@TMZ/Eud-Cs. Figure 5A depicts the composite sample of Se@TMZ/Eud-Cs. All the index peaks of ingredients as SNPs, TMZ, Cs, and Eud can be seen in this figure. It can be attributed to stretching and bending vibration of Se–O at 1105 and 470 cm^−1^, respectively, as they can be observed in Figure 5B. O–H stretch from 3300–2500 cm^−1^ and C=O stretch from 1760–1690 cm^−1^ confirmed the presence of carboxylic acid functional groups in Eud (Figure 5C). Those stretching vibrations were pointed to Eud presence in the Se@TMZ/Eud-Cs composition. In addition, the functional indicator groups of Cs as C–N and C=O are visible in the absorption spectra of stretching vibration at 1451 and 1733 cm^−1^, respectively (Figure 5D). In the TMZ spectrum, the N=N vibration peak at 1595 cm^−1^ appears in Figure 5E. The structure of temozolomide in Se@TMZ/Eud-Cs can be detected by FTIR by N=N stretching vibration at 1471 cm^−1^, which decreased due to the presence of metal complexes.

#### 3.1.5. TGA Analysis

The modified surface of Se@TMZ/Eud-Cs was further confirmed by TGA analysis under a constant N_2_ flow (Figure 6). Results showed that Se@TMZ/Eud-Cs and Se@TMZ/Eud lost up to 56.45% and 67.95% of their total mass at 798.1 °C, while Se@TMZ and SNPs only lost up to 68.3% and 94.50% of their mass at the same temperature, respectively. The low weight loss in the thermogram of SNPs indicated the evaporation of the residual solvent, adsorbed water, and NO-functional group on these nanoparticles. The thermogravimetry of Se@TMZ/Eud-Cs, Se@TMZ/Eud, and Se@TMZ had higher carbon contents than SNPs related to the high drug loading of TMZ. TMZ loading in the nanoparticles was influenced by different factors, such as nanoparticle surface charge, electrostatic attraction between TMZ, SNPs, and drug encapsulation by Eud and Cs-coated particles. These factors affected Se@TMZ/Eud-Cs to get high payload as shown in Figure 6. While Se@TMZ/Eud showed a lower TMZ loading than Se@TMZ/Eud-Cs, it could be a point to the importance of Cs for drug capture in nanoparticles. The difference in weight loss between Se@TMZ/Eud and Se@TMZ/Eud-Cs, and between Se@TMZ and Se@TMZ/Eud can be a hint to the decomposable nature of Eud and Cs layer, respectively.

#### 3.1.6. Drug Loading and In Vitro Release of Nanoparticles

TMZ loading in the nanoparticles was influenced by different factors, such as nanoparticle surface charge, electrostatic attraction between TMZ, SNPs, and drug encapsulation by Eud and Cs-coated particles. These factors affected Se@TMZ/Eud-Cs to get high payload as shown in Table 2. While Se@TMZ/Eud showed a lower TMZ loading than Se@TMZ/Eud-Cs, it could be a point to the importance of Cs for drug capture in nanoparticles.

The in vitro release profiles of TMZ from Se@TMZ/Eud and Se@TMZ/Eud-Cs in two buffer media with pH = 5 and 7.4 are shown in Figure 7. It was detected that drug release was pH-dependent because of the significantly higher TMZ release at pH = 5 compared to pH = 7.4 (*p*-value < 0.0001). Se@TMZ/Eud release profile showed that TMZ was burst-released in the first 5 h and faster than Se@TMZ/Eud-Cs. While Se@TMZ/Eud-Cs prevented burst release significantly in the early hours due to Cs coating (*p*-value < 0.0001). Therefore, in view of these observations, the cancer cells have an acidic pH [37]; this drug delivery system has helped targeted TMZ delivery to brain cancer cells due to pH sensitivity of the nanoparticles and TMZ can be released in a predictable and required manner (Figure 7).

#### 3.1.7. Cellular Uptake

TMZ has no measurable autofluorescence activity [38]. Therefore, the amount of cellular uptake was measured using labeled Se@Cur/Eud-Cs with curcumin [31]. The results showed a significantly higher uptake of Se@Cur/Eud-Cs compared to free Cur (*p*-value < 0.01) in C6 cells after 4 h with 90.35 ± 1.65 and 72.05 ± 1.95 rates of positive fluorescent cells, respectively (Figure 8).

### 3.2. Cytotoxicity Studies

Free TMZ showed a dose-dependent inhibitory effect on the viability of C6 cells after 48 h treatment, which was same as SNP loaded with TMZ. According to the MTT assay results, Se@TMZ/Eud-Cs and Se@TMZ/Eud were dramatically more effective on increasing cytotoxicity (*p*-value < 0.0001) with significantly lower IC_50_ value (205.76 ± 8.69 and 579.26 ± 6.47, respectively) compared with free TMZ (918.50 ± 8.64) (Figure 9).

### 3.3. Apoptosis Assay

As depicted in Figure 10, although early apoptosis was induced significantly in both of the cell groups treated with coated SNPs containing 200 μM TMZ (*p*-value < 0.0001) and free TMZ with the same concentration (*p*-value < 0.01) compared to the control, late apoptosis was increased just by the influence of the nanoparticles (*p*-value < 0.0001). Moreover, a comparison of free TMZ and Se@TMZ/Eud-Cs showed notably higher induction of early and late apoptosis (*p*-value < 0.0001). However, in cells treated with Se@TMZ/Eud, no significant difference was seen in the induction of early apoptosis (Figure 10 and Table 3).

### 3.4. Gene Expression Level Assay

Gene expression levels of MGMT, E2F6, and RELA genes after 48 h of treatment with TMZ, Se@TMZ-Eud, and Se@TMZ-Eud-Cs showed that MGMT mRNA level increased in TMZ treated cells significantly (*p* < 0.05) compared to the untreated group. Treatment with Se@TMZ-Eud and Se@TMZ-Eud-Cs decreased MGMT notably (*p* < 0.05 and *p* < 0.01 respectively). Cs had more influence on the reduction of the MGMT expression. RELA mRNA level increased in TMZ-treated cells compared to control cells. RELA mRNA level reduced in both Se@TMZ-Eud- and Se@TMZ-Eud-Cs-treated groups significantly (*p* < 0.05) compared with the TMZ treated group. E2F6 mRNA level remarkably increased with Se@TMZ-Eud and decreased with Se@TMZ-Eud-Cs compared to the TMZ-treated group (*p* < 0.05) (Figure 11). 

## 4. Discussion

Resistance to TMZ is a major cause of recurrence of GBM, resulting in high mortality of GBM worldwide [2,8]. The molecular mechanisms involved in TMZ resistance in GBM patients have been described [39,40]. Therefore, the effectiveness of anti-cancer drugs could be improved by utilizing approaches to control these molecular factors. Delivering the effective concentrations of therapeutic agents to tumors is a strategic factor in increasing cancer treatment efficiency. Today, nanocarriers are used as potential vectors in delivering agents to cancers, such as GBM [41,42]. In this study, we used Se@TMZ/Eud-Cs nanoparticles to deliver TMZ to C6 cells.

SNPs were prepared by precipitation method and coated with Eud and Cs after TMZ loading for targeted brain delivery in the C6 cell line. Formulated Se@TMZ/Eud-Cs were investigated for intracellular delivery of TMZ and expression of MGMT, E2F6, and RELA genes in C6 cells. Monodispersed SNPs with a particle size of about 200 nm were successfully prepared. As showed in Figure 2A SNPs revealed spherical shape and narrow size distribution. After coating the SNPs with Eud and Cs, Se@TMZ/Eud and Se@TMZ/Eud-Cs displayed a thin layer on the nanoparticle surface, which increased the particle size (Figure 2B,C). As was reported in several studies, chemical preparation of SNPs can produce spherical SNPs in the size range of 30–200 nm [43,44,45]. Moreover, DLS results with SLB calculation confirmed the SEM data for the SNPs and Se@TMZ/Eud-Cs. As depicted in Figure 3a,b SNPs and Se@TMZ/Eud-Cs displayed sharp peaks at 169 and 228 nm, respectively. This is also in agreement with the SEM results and attests a narrow size distribution. Zeta potential of empty SNPs was measured to be −20 mV [46], which increased to +32 mV due to the adsorption of positively charged Cs. A previous study by Daryasari et al. showed the same results for a different type of nanoparticle coated with Cs [31]. 

EDAX analysis of the Se@TMZ/Eud-Cs confirmed surface modifications of SNPs with different compositions of different molecules. As shown in Figure 5 the selenium, carbon, oxygen, nitrogen, and chlorine peaks were found with the atomic ratios of 0.1, 48.4, 36.1, 13.8, and 0.7 (%), respectively. Carbon, oxygen, nitrogen elements are usually found in most compounds such as Cs, Eud, and TMZ. Appearance of a chlorine peak in the EDAX spectrum could be proof that Eud was located on the surface of Se@TMZ/Eud-Cs.

As pointed out in the Results section, the structure of Se@TMZ/Eud-Cs was confirmed by the FTIR method. The FTIR spectrum of Se@TMZ/Eud-Cs was compared with the free compound as Cs, Eud, and TMZ. Figure 5A showed the composite sample of Se@TMZ/Eud-Cs. In Figure 5B, the results of SNPs studies by FTIR are shown. However, some peaks including C=O and Se-O were detected. These data attest the existence of some entrapment of started material, e.g., SeO_2_ and ascorbic acid by the SNPs [37]. In Figure 5C,D, results of Eud and Cs analysis are depicted. All the index peaks for Eud and Cs as C=O and C=O stretching, respectively, were found in the absorption spectra of Se@TMZ/Eud-Cs. Furthermore, the spectrum of TMZ was identified by N=N vibration peak at 1595 cm^−1^ (Figure 5E). It can be detected by N=N stretching vibration at 1471 cm^−1^ in Se@TMZ/Eud-Cs, which was reduced by the presence of metal complexes [47].

TGA was used for further analysis of the final structure of Se@TMZ/Eud-Cs and its compositions in the later steps. As mentioned before, SNP only lost up to 94.50% of its mass at 798.1 °C. This low mass reduction is related to the evaporation of the residual solvent and adsorbed water. SNPs do not have any functional groups, however, as pointed above, a little residue of SeO_2_ and ascorbic acid in SNPs were burned here. A sharp decrease of Se@TMZ/Eud-Cs, Se@TMZ/Eud and Se@TMZ mass between 70 to 130 °C was indicative of high loading of TMZ in these nanoparticles. This assumption could be supported by the fact that the boiling point of TMZ was detected to be about 212 °C. Therefore, TMZ was evaporated gradually till 210 °C and after that, all the three formulations of loaded TMZ nanoparticles did not show any mass decrease. Burning of organic material happened in the temperature range of 300–650 °C. Consequently, the removal of Cs and Eud should be seen in this range. The second sharp decline in Se@TMZ/Eud-Cs and Se@TMZ/Eud masses were confirmed by the elimination of Cs and Eud compositions from these formulations. The difference in the mass decline between Se@TMZ/Eud-Cs and Se@TMZ/Eud was an evidence of the extra layer of Cs on Se@TMZ/Eud-Cs.

Most of the studies related to the TMZ stability deal with the chemical modification of this drug. This demonstrates that the prepared polymeric nanoparticles were able to protect TMZ from hydrolysis. The significant impact of the nanoparticles in preserving TMZ from hydrolysis is evident. Free TMZ is quickly hydrolyzed to the inactive metabolites during the time between the sample preparation and the contact with the media, which leads to its inactivation [10]. Our cellular uptake results suggest that Cs-coated nanoparticles improved targeted delivery, followed by an increase in cellular uptake and cytotoxicity of this drug. Cells treated with SNPs loaded with TMZ had an upper death potential compared to those treated with TMZ, especially in treatment with higher concentrations. A recent study demonstrated that the combination of Cs, biotin, and TMZ can enhance the biological effects of TMZ [48]. In our study, similarly, the Cs compound decreased IC_50_ and increased cell apoptosis compared to TMZ in the C6 cells. This could be due to the ability of Cs to interact with cell surface macromolecules and increase cellular uptake. Cs is widely used in drug nanocarriers to improve drug loading, sustained release, cellular uptake, and drug targeting. Furthermore, the positive charge of Cs in the nanoparticles is effective in adsorption to target cells [49].

The formulated Cs nanoparticles demonstrated acceptable results in the treatment of some brain diseases, such as glioma, Parkinson’s, and Alzheimer’s [50]. Se@TMZ/Eud-Cs significantly improved the stability of TMZ at acidic and neutral pH by reducing IC_50_. By the treatment of the C6 cells with Se@TMZ/Eud and Se@TMZ/Eud-Cs, after 48 h, the expression of main genes in drug resistance, such as MGMT and RELA, was decreased significantly compared with TMZ treatment. Treatment with TMZ leads to MGMT over-expression. Other researcher groups showed similar up-regulation [51]. Our study showed that presence of Cs in the nanodrugs is more effective in reducing gene expression compared with the formulations with no Cs. The decreased expression may lead to decreased therapeutic resistance. Therefore, it seems that nanodrugs will have a significant effect on reducing resistance by affecting their signaling pathways. However, more studies are needed to investigate the effect of our nanodrug on TMZ resistance. Regarding E2F6 expression, an increase in gene expression was observed for Se@TMZ/Eud formulation. Although E2F6 expression for treatment with Se@TMZ/Eud-Cs was reduced similarly to MGMT and RELA, it is possible that the presence of Cs cells was effective on the reduction of the E2F6 expression.

One of the known mechanisms of inducing TMZ resistance by MGMT is through RELA. RELA has a role in inducing MGMT gene expression [52]. DNA damage by TMZ activates NF-κB. Therefore, significant increase in E2F6 expression and promoting GBM cell survival leads to TMZ resistance. Previous studies have shown that E2F6, E2F7, and E2F8 play a key role in repairing damaged DNA in cells treated with a DNA degradation agent and increase resistance to chemotherapy drugs and cell survival. These findings suggest that inhibition of E2F6 is a strategy for the development of therapies. On the other hand, evaluating E2F6 expression can be useful as a biomarker. Research findings indicate that suppression of E2F6 by inactivating NF-κB can increase TMZ sensitivity [4]. One study showed that prolonged exposure to TMZ might increase resistance and the malignant phenotype in the malignant glioma cells [51]. Similarly, our gene expression analysis showed that TMZ increased MGMT gene expression level after 48 h. Another study reported that increased activity of the PI3K/AKT signaling pathway in GBM-induced TMZ resistance because this increased activity caused the NF-κB p65 transcript to increase the expression of some genes, such as MGMT [52]. Another study reported that the signaling pathway (MEK)-(ERK)-(MDM2)-p53 is involved in the MGMT gene expression [53]. Authors postulated that the inhibition of MEK and ERK activates P53 and reduces the MGMT gene expression and leads to induce sensitivity of glioma cells to TMZ. The findings indicate that targeting the MEK-ERK-MDM2-p53 pathway in combination with TMZ could be a new treatment and a promising strategy in GBM treatment [53]. On the other hand, NF-κB is known as an important factor in MGMT transcription independent of the MGMT methylation status. It has previously been shown that a high level of NF-κB promotes GBM alkylating agent-based chemotherapy resistance. This resistance is associated with the activation of gene transcription. The results suggest that cell exposure to the alkylating agent activates NF-κB and increased MGMT level. Therefore, NF-κB inhibitors may help to overcome the chemical resistance induced by alkylating chemotherapy [54]. There is a direct relationship between NF-κB/p65 and MGMT. There are two NF-κB binding sites in the MGMT promoter region, which may indicate that NF-κB is an important transcription factor in the regulation of MGMT transcription. Inhibition of NF-κB can reduce MGMT gene expression [52]. Therefore, NF-κB/MGMT pathway regulation appears to be a new treatment strategy to overcome TMZ resistance in GBM. On the other hand, the NF-κB pathway is also involved in inhibiting apoptosis. GBM cells express high levels of BCL2 anti-apoptotic family proteins, such as Bcl-2 and Bcl-xL, which may make these cells resistant to apoptosis. Inhibition of NF-κB expression can inhibit Bcl-2 expression and enhance Bax expression, thereby causing apoptosis [49]. The gene expression and apoptosis results showed that treatment by nanoparticles including Cs has acceptable effects. One of the reasons for the increase in the antitumor activity of Cs-containing nanoparticles is its positive charge and stability in the acidic environment of tumors.

## 5. Conclusions

Characterization of the Se@TMZ/Eud-Cs formulations showed that our designed nanodrug properties are promising. Its cellular uptake is suitable for drug delivery applications. Compared to the free form of the drug (TMZ), Se@TMZ/Eud-Cs formulation is more efficient in inducing cytotoxicity, apoptosis, and regulation of resistance-related genes, including MGMT, RELA, and E2F6. Therefore, it seems that using a suitable nanocarrier, such as Se@TMZ/Eud-Cs, can be effective in increasing anti-cancer effects and reducing therapeutic resistance of the conventional anti-cancer agents such as TMZ for GBM treatment in the future.

## Figures and Tables

**Figure 1 nanomaterials-11-01704-f001:**
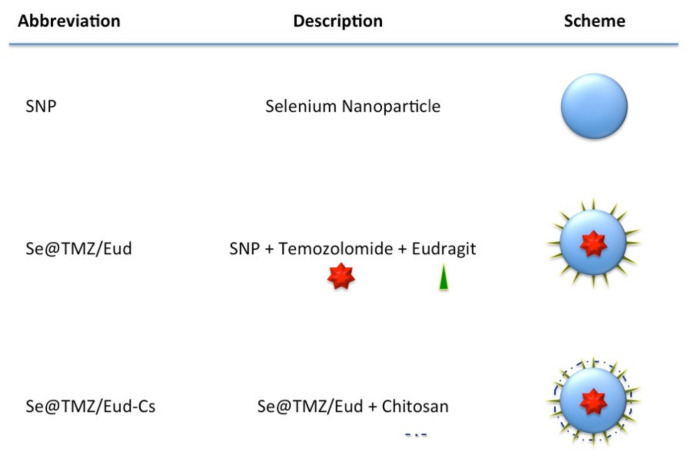
Designations and schemes of the formulations designed in current study. The work flow of the synthesis of nanoparticles and their characterization can be summarized as follows: In the first step of nanoparticle preparation, precipitation of selenium by ascorbic acid solution of selenium dioxide was performed and the selenium nanoparticles (SNPs) were constructed. Surface adsorption of TMZ on the selenium nanoparticles was achieved in the next step. Precipitation of Eud on Se@TMZ was performed and then Se@TMZ/Eud was coated by chitosan (Cs) to prepare Se@TMZ/Eud-Cs. Thereafter, characterization of Se@TMZ/Eud-Cs including morphology, size, zeta potential, elemental composition, chemical structure and in vitro drug loading and release were carried out.”-.-“ represents chitosan presence on the particles.

**Figure 2 nanomaterials-11-01704-f002:**
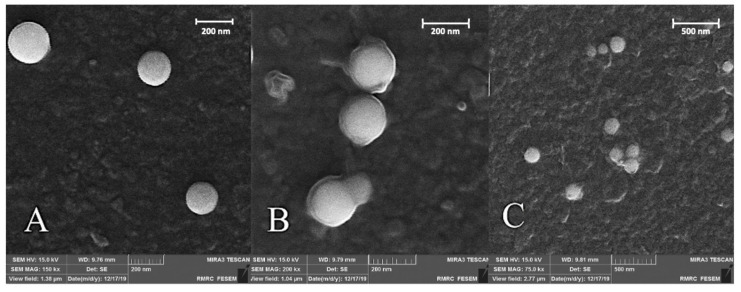
FE-SEM images of (**A**) SNPs, (**B**) Se@TMZ/Eud, and (**C**) Se@TMZ/Eud-Cs.

**Figure 3 nanomaterials-11-01704-f003:**
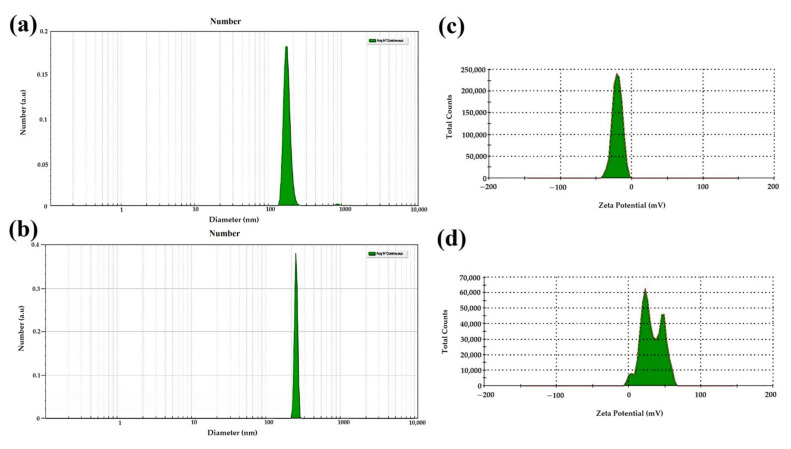
Size distribution and zeta potential were obtained by using DLS and SBL. (**a**) DLS evaluation of the number of SNPs, (**b**) DLS evaluation of the number of Se@TMZ/Eud-Cs, (**c**) Zeta potential of SNPs, (**d**) Zeta potential of Se@TMZ/Eud-Cs.

**Figure 4 nanomaterials-11-01704-f004:**
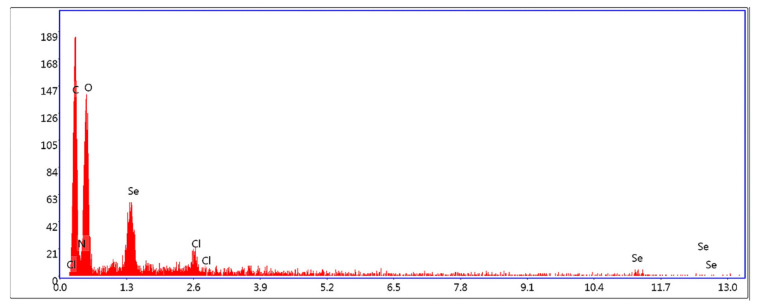
EDAX spectra obtained for Se@TMZ/Eud-Cs.

**Figure 5 nanomaterials-11-01704-f005:**
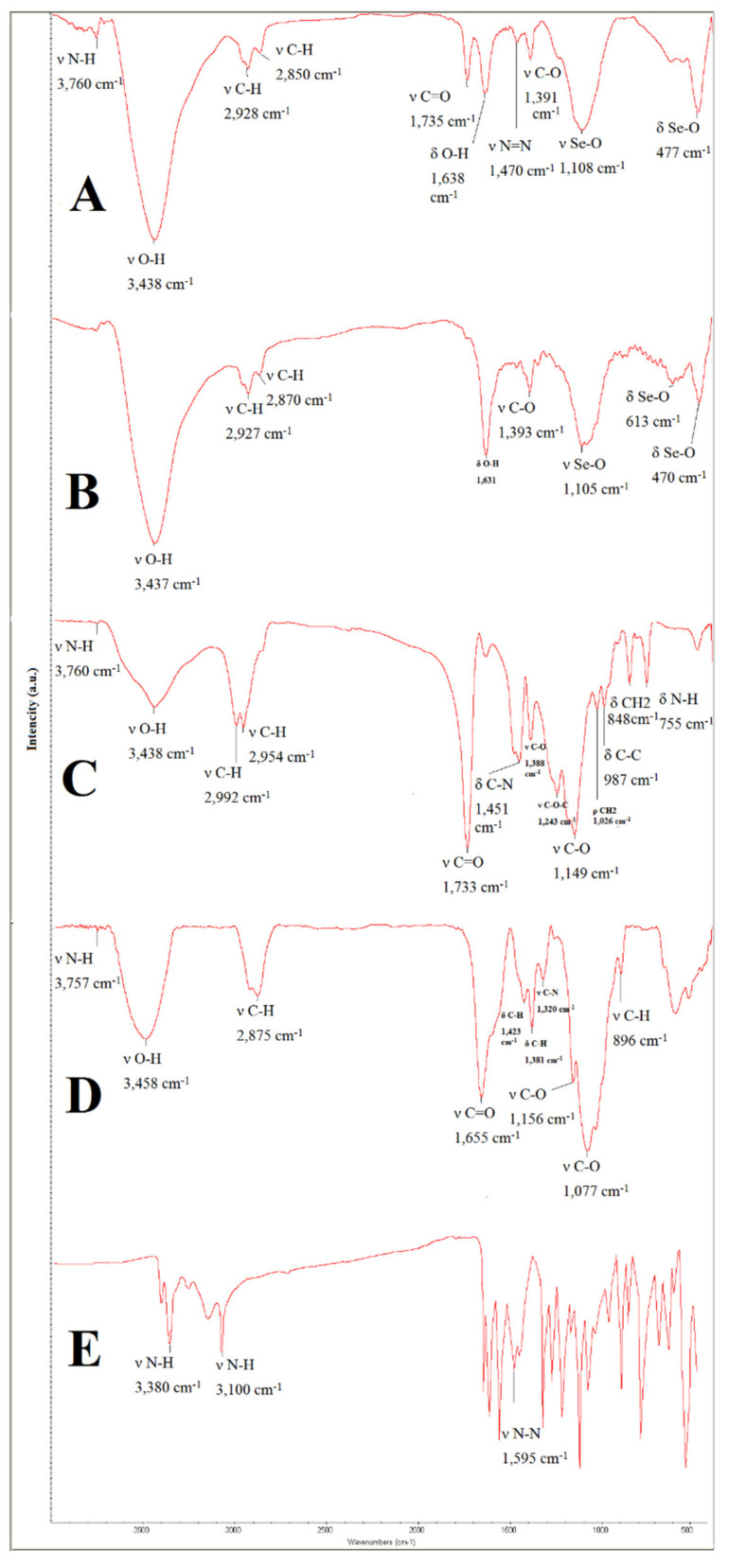
FTIR spectra of: (**A**) Se@TMZ/Eud-Cs, (**B**) SNPs, (**C**) Eud, (**D**) Cs, and (**E**) TMZ.

**Figure 6 nanomaterials-11-01704-f006:**
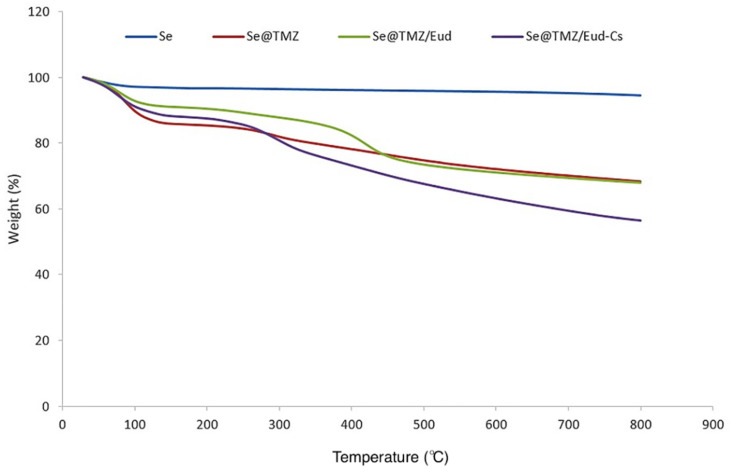
TGA analysis of the formulated nanoparticles under a constant N_2_ flow. Se@TMZ/Eud-Cs and Se@TMZ/Eud lost up to 56.45% and 67.95% of their total mass at 798.1 °C, while Se@TMZ and SNPs only lost up to 68.3% and 94.50% of their mass at the same temperature, respectively.

**Figure 7 nanomaterials-11-01704-f007:**
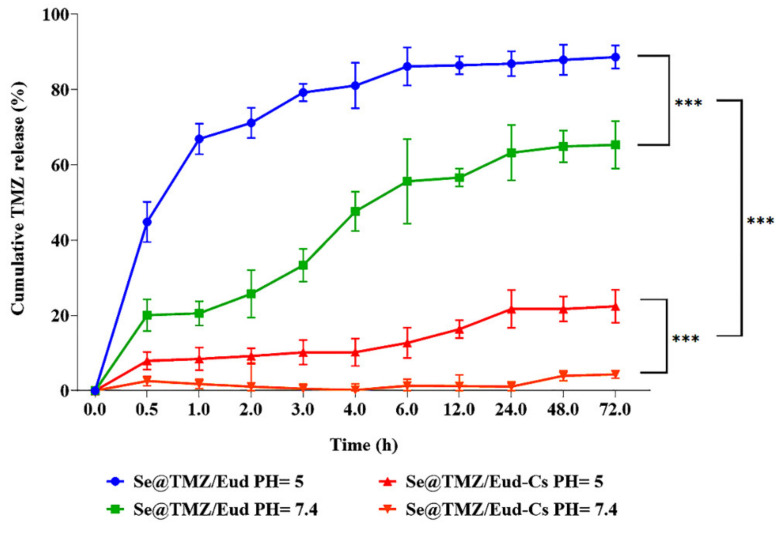
In vitro TMZ release profile from Se@TMZ/Eud and Se@TMZ/Eud-Cs at pH = 5 and 7.4. Both coated SNPs loaded with TMZ released significantly more concentration of drug at pH 5 compared to pH 7.4 (*p*-value < 0.0001), indicating pH-dependence of drug-release. Moreover, at all specified time points, Se@TMZ/Eud-Cs exhibited a dramatically higher cumulative release of TMZ than Se@TMZ/Eud-Cs (*p*-value < 0.0001). Data are represented as mean ± SD, *n* = 3. *** *p*-value < 0.0001.

**Figure 8 nanomaterials-11-01704-f008:**
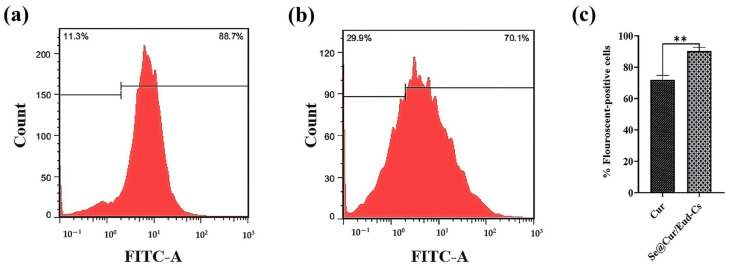
Cellular uptake of SNPs. Curcumin (Cur) molecule, which has intrinsic fluorescent properties, was used to label SNPs. For the synthesis of Se@Cur/Eud-Cs, 5 mL of Cur solution (1 mg/mL in acetone) was used instead of TMZ solution and cellular uptake of Cur and Se@Cur/Eud-Cs was evaluated by flow cytometry. The results showed higher uptake of (**a**) Se@Cur/Eud-Cs rather than (**b**) free Cur (*p*-value < 0.01) in C6 rat glioblastoma cells after 4 h treatment. (**c**) Cellular uptake of Se@Cur/Eud-Cs compared to free Cur in the C6 cells after 4 h treatment. Data are represented as mean ± SD, *n* = 3. ** *p*-value < 0.01.

**Figure 9 nanomaterials-11-01704-f009:**
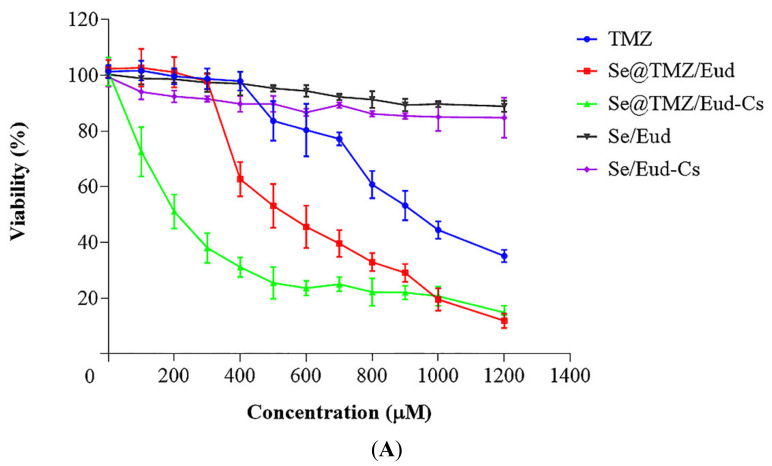
(**A**,**B**) Two different representation of the effects of Free TMZ in comparison with Se@TMZ/Eud and Se@TMZ/Eud-Cs loaded with the same concentration of TMZ on the cell cytotoxicity of C6 rat glioblastoma cells after 48 h. The cells were treated with different concentrations of the samples for 48 h. The IC50 value of TMZ, Se@TMZ/Eud, and Se@TMZ/Eud-Cs were 918.50 ± 8.64, 579.26 ± 6.47 and 205.76 ± 8.69 μM, respectively. Data are represented as mean ± SD, *n* = 3.

**Figure 10 nanomaterials-11-01704-f010:**
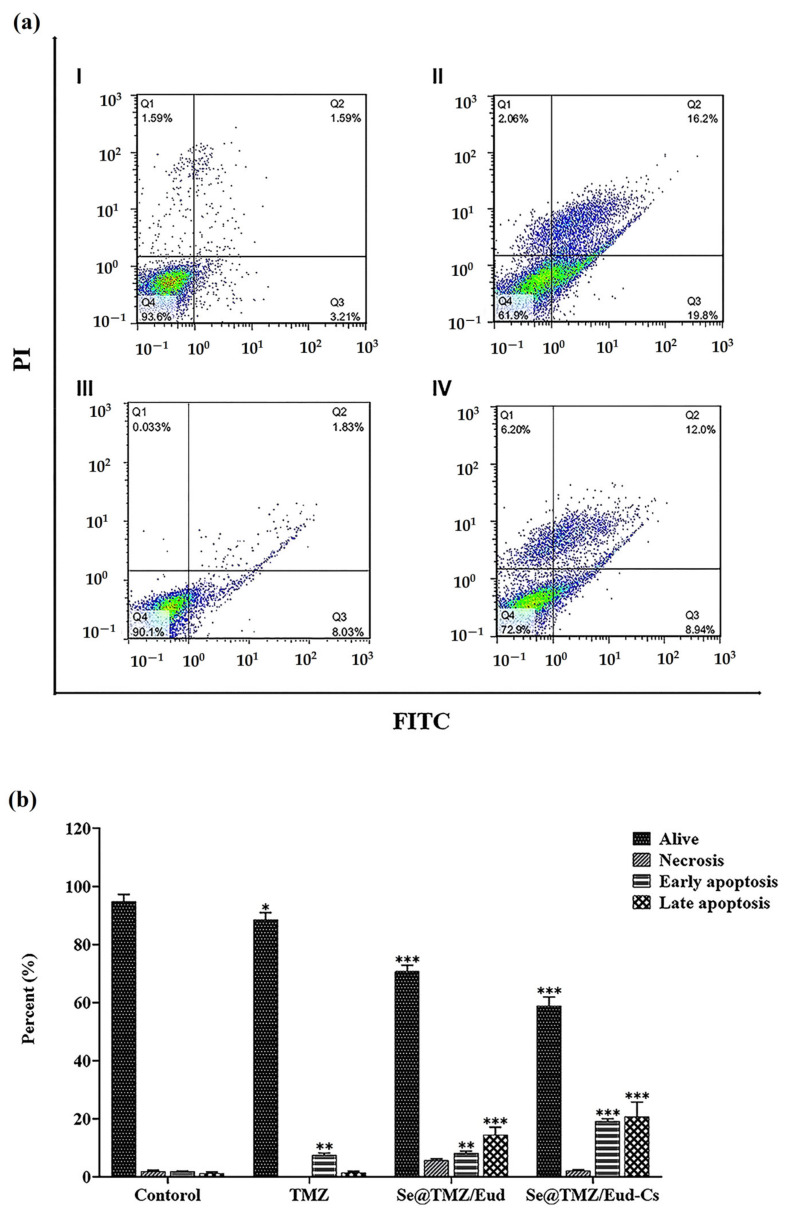
Effect of TMZ, Se@Cur/Eud, and Se@Cur/Eud-Cs on cell apoptosis. Untreated cells were employed as a control. (**a**) Stained cells (Q1, AnnexinV +/P I−), (Q2, AnnexinV +/PI +), (Q3, AnnexinV −/PI +), and (Q4, AnnexinV −/PI −) are identified as necrotic, delayed-phase of apoptosis, primary-phase of apoptosis, and alive cells, respectively. (**b**) SNPs loaded with TMZ induced significantly higher apoptosis than free TMZ (*p*-value < 0.0001) compared to the control. Date expressed as mean ± SD, *n* = 3. * *p* value < 0.05, ** *p* value < 0.01, *** *p* value < 0.0001.

**Figure 11 nanomaterials-11-01704-f011:**
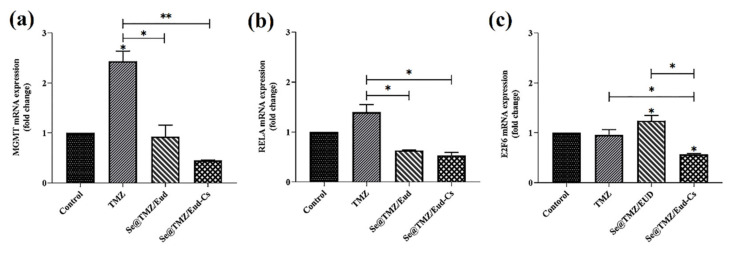
Expression levels of MGMT, E2F6, and RELA genes. (**a**) MGMT mRNA level significantly increased in TMZ treatment cells vs. untreated group. Treatment with Se@TMZ-Eud and Se@TMZ-Eud-Cs decreased MGMT significantly. (**b**) RELA mRNA level significantly reduced in Se@TMZ-Eud- and Se@TMZ-Eud-Cs-treated groups vs. TMZ-treated group. (**c**) E2F6 mRNA level significantly increased with Se@TMZ-Eud and decreased with Se@TMZ-Eud-Cs vs. TMZ-treated group. Data are presented as mean ± SD, *n* = 3. * *p* < 0.05 and ** *p* < 0.01.

**Table 1 nanomaterials-11-01704-t001:** The sequence of Real-Time PCR primers used in the study.

Gene Name	Forward Primer (5′–3′)	Reverse Primer (5′–3′)	Annealing Tm (°C)	PCR Product Length
MGMT	GACAGGTGTTATGGAAGCTG	GGCTGCTAATTGCTGGTAAGA	58	71
E2F6	ATGTGTCGCTGGTCTACTTA	ATGTGGTTCTTAGATTTCTTTTCC	56	178
RELA	CGTGAGGCTGTTTGGTTTGAG	GTCTTATGGCTGAGGTCTGGTC	62	109
Actb	AGAGGGAAATCGTGCGTGAC	AGGAAGGAAGGCTGGAAGAGA	60	187

**Table 2 nanomaterials-11-01704-t002:** TMZ loading profile. Data are represented as mean ± SD, *n* = 3.

Nanoparticles	TMZ Loading Efficiency (%)	TMZ Loading Capacity (%)
Se@TMZ/Eud-Cs	82.77 ± 5.30	37.5 ± 4.43
Se@TMZ/Eud	61.70 ± 3.65	15.71 ± 3.43

**Table 3 nanomaterials-11-01704-t003:** Cell viability and death percentage in different cell groups.

	Live Cells (%)	Necrotic Cells (%)	Early Apoptotic Cells (%)	Late Apoptotic Cells (%)
Control	94.91 ± 2.39	1.96 ± 0.35	1.8 ± 0.24	1.14 ± 0.57
Free TMZ	88.63 ± 2.43	0.06 ± 0.03	7.54 ± 0.65	1.41 ± 0.44
Se@TMZ/Eud	70.72 ± 2.20	5.73 ± 0.50	8.18 ± 0.68	14.52 ± 2.63
Se@TMZ/Eud-Cs	58.86 ± 3.16	2.18 ± 0.27	19.16 ± 0.85	22.88 ± 4.98

## Data Availability

The data that support the findings of this study are available from the corresponding author upon reasonable request.

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
