# Peer review of "Anticancer Potential of Temozolomide-Loaded Eudragit-Chitosan Coated Selenium Nanoparticles: In Vitro Evaluation of Cytotoxicity, Apoptosis and Gene Regulation"

_nanomaterials, 2021, doi:10.3390/nano11071704_

Round 1

Reviewer 1 Report

The nanoparticles temozolomide-loaded with-Chitosan are well characterised and present interesting data related to the cytotoxicity, apoptosis and gene regulation.

Additional work flow of the nanoparticles synthesis and characterisation.

Please define clear: (A) SNPs (B) Se@TMZ/Eud and (C) Se@TMZ/Eud-Cs.

Please define the negative and positive control used in the experiment.

Line 397- reference error

Line  224-234 - please rephrase

 Line 424 " which increased the particle size ("

Author Response

Please find below our answers to reviewers for the manuscript (Original, Research Article) titled:

“Anticancer Potential of temozolomide-loaded Eudragit-Chitosan coated selenium nanoparticles: in vitro evaluation of cytotoxicity, apoptosis and gene regulation”

REVIEWER 1:

We thank the Reviewer for constructive comments and suggestions which we have addressed in details as follows:

The nanoparticles temozolomide-loaded with-Chitosan are well characterised and present interesting data related to the cytotoxicity, apoptosis and gene regulation.

Encouraging statement is much appreciated.

Additional work flow of the nanoparticles synthesis and characterisation.

This is now added in the Legend to Figure 1 that is added based on the next suggestion of the Reviewer.

Please define clear: (A) SNPs (B) Se@TMZ/Eud and (C) Se@TMZ/Eud-Cs.

A new Figure is added (Figure 1) in comply with the suggestion of Reviewer.

Please define the negative and positive control used in the experiment.

In comply with the Reviewer, we have replaced Figure 9 and converted a single Figure into two sections (Fig. 9A and 9B) to indicate controls used more clearly. In this study we have used the free (un-encapsulated) drug (TMZ) as a control. Additional controls were: selenium-eudragit (Se/Eud), selenium-eudragit-chitosan (Se/Eud-Cs), and selenium-TMZ-eudragit (Se@TMZ/Eud).

The controls used in our study are in agreement with those employed in a similar study:

“Investigation of Dextran-Coated Superparamagnetic Nanoparticles for Targeted Vinblastine Controlled Release, Delivery, Apoptosis Induction, and Gene Expression in Pancreatic Cancer Cells”

https://doi.org/10.3390/molecules25204721

Line 397- reference error

The error was originated due to conversion of our manuscript file by Journal and do not appear in the file visible to us.  

Line 224-234 - please rephrase

The section is now rephrased in agreement with the suggestion of Reviewer.

Line 424 " which increased the particle size ("

The error was originated due to conversion of our manuscript file by Journal and do not appear in the file visible to us. 

Reviewer 2 Report

The authors demonstrated the anticancer activity of Se@TMZ/Eud-Cs in glioblastoma cells. Overall, the results sound interesting. However, several important experiments are missing, and some points should be revised as shown below.

  1. The authors did not suggest any molecular mechanism of Se@TMZ-Eud-Cs-induced apoptosis. Is it the same with TMZ-induced apoptosis? More experiment should be conducted to determine the possible target of Se@TMZ-Eud-Cs .
  2. The authors suggest the drug resistance-overcoming effect of Se@TMZ-Eud-Cs. However, they just showed the mRNA level of TMZ resistance-related genes. They should show more evidence that Se@TMZ-Eud-Cs can suppress TMZ resistance.
  3. The activity (phosphorylation level) of NF-kB is more important than the mRNA level of RelA. Please investigate the activity of NF-kB after treatment of each drug.
  4. All abbreviations should be defined at their first appearance in Abstract, main text, and Figure legends. For example, GBM was not defined in line 46.
  5. In Results section, the subtitle of each result should be the conclusion of each Figure, not the name of each assay. In addition, the authors just describe the Figures without explaining the meaning of the results. I think much part of the Discussion section describing the meaning of each result should be rearranged to the Results section. This rearrangement can make the Discussion clear and concise.

Author Response

REVIEWER 2:

  1. The authors did not suggest any molecular mechanism of Se@TMZ-Eud-Cs-induced apoptosis. Is it the same with TMZ-induced apoptosis? More experiment should be conducted to determine the possible target of Se@TMZ-Eud-Cs.

The mechanism of action of the antineoplastic drug used in this study (i.e., temozolomide) is mentioned in the manuscript. Furthermore, the role of encapsulation of conventional anticancer therapeutics is commonly known with respect of increasing shelf-life of the drug product and providing targeting / theranostic effects.   

We thank Reviewer for this highly constructive suggestion, which bestows a completely separate study and is out of the scope of current study.

  1. The authors suggest the drug resistance-overcoming effect of Se@TMZ-Eud-Cs. However, they just showed the mRNA level of TMZ resistance-related genes. They should show more evidence that Se@TMZ-Eud-Cs can suppress TMZ resistance.

With respect of this comment of Reviewer, the manuscript mentions (under Discussion):

“The formulated Cs nanoparticles demonstrated acceptable results in the treatment of some brain diseases, such as glioma, Parkinson's, and Alzheimer's [50]. Se@TMZ/Eud-Cs significantly improved the stability of TMZ at acidic and neutral pH by reducing IC50. By the treatment of the C6 cells with Se@TMZ/Eud and Se@TMZ/Eud-Cs, after 48 hours, the expression of main genes in drug resistance, such as MGMT and RELA was decreased significantly compared with TMZ treatment. Treatment with TMZ leads to MGMT over-expression. Other researcher groups showed similar up-regulation [51]. Our study showed that presence of Cs in the nano-drugs is more effective in reducing gene expression compared with the formulations with no Cs. The decreased expression may lead to decreased therapeutic resistance. Therefore, it seems that nano-drugs will have a significant effect on reducing resistance by affecting their signaling pathways. However, more studies are needed to investigate the effect of our nano-drug on TMZ resistance.

In comply with the technical suggestion of Reviewer, we will consider further elucidation of the mentioned mechanisms in our subsequent studies.

  1. The activity (phosphorylation level) of NF-kB is more important than the mRNA level of RelA. Please investigate the activity of NF-kB after treatment of each drug.

We thank Reviewer for this expert suggestion and will consider investigating the activity of NF-kB after treatment of each drug in our future studies.

  1. All abbreviations should be defined at their first appearance in Abstract, main text, and Figure legends. For example, GBM was not defined in line 46.

Corrections have been done in comply with the suggestion of Reviewer.

  1. In Results section, the subtitle of each result should be the conclusion of each Figure, not the name of each assay. In addition, the authors just describe the Figures without explaining the meaning of the results. I think much part of the Discussion section describing the meaning of each result should be rearranged to the Results section. This rearrangement can make the Discussion clear and concise.

We appreciate the constructive suggestions of this Reviewer. Changes have now done accordingly.

Round 2

Reviewer 1 Report

No further correction/

Reviewer 2 Report

Overall, the manuscript was revised according to the reviewer's suggestion.

Hope the authors conduct further experiments as suggested by the reviewer in their subsequent studies.